# Altered Surface Hydrophilicity on Copolymer Scaffolds Stimulate the Osteogenic Differentiation of Human Mesenchymal Stem Cells

**DOI:** 10.3390/polym12071453

**Published:** 2020-06-29

**Authors:** Zhe Xing, Jiazheng Cai, Yang Sun, Mengnan Cao, Yi Li, Ying Xue, Anna Finne-Wistrand, Mustafa Kamal

**Affiliations:** 1School of Stomatology, Lanzhou University, Lanzhou 730000, China; xingz@lzu.edu.cn (Z.X.); caijzh19@lzu.edu.cn (J.C.); caomn16@lzu.edu.cn (M.C.); 2Department of Clinical Dentistry, Faculty of Medicine, University of Bergen, 5009 Bergen, Norway; kamal.mustafa@uib.no; 3Department of Fibre and Polymer Technology, Royal Institute of Technology (KTH), SE-100 44 Stockholm, Sweden; jimix_sun@hotmail.com (Y.S.); annaf@kth.se (A.F.-W.)

**Keywords:** bone marrow mesenchymal stem cells, copolymer, Tween 80, scaffold, hydrophilicity

## Abstract

Background: Recent studies have suggested that both poly(l-lactide-*co*-1,5-dioxepan-2-one) (or poly(LLA-*co*-DXO)) and poly(l-lactide-*co*-ε-caprolactone) (or poly(LLA-*co*-CL)) porous scaffolds are good candidates for use as biodegradable scaffold materials in the field of tissue engineering; meanwhile, their surface properties, such as hydrophilicity, need to be further improved. Methods: We applied several different concentrations of the surfactant Tween 80 to tune the hydrophilicity of both materials. Moreover, the modification was applied not only in the form of solid scaffold as a film but also a porous scaffold. To investigate the potential application for tissue engineering, human bone marrow mesenchymal stem cells (hMSCs) were chosen to test the effect of hydrophilicity on cell attachment, proliferation, and differentiation. First, the cellular cytotoxicity of the extracted medium from modified scaffolds was investigated on HaCaT cells. Then, hMSCs were seeded on the scaffolds or films to evaluate cell attachment, proliferation, and osteogenic differentiation. The results indicated a significant increasing of wettability with the addition of Tween 80, and the hMSCs showed delayed attachment and spreading. PCR results indicated that the differentiation of hMSCs was stimulated, and several osteogenesis related genes were up-regulated in the 3% Tween 80 group. Poly(LLA-*co*-CL) with 3% Tween 80 showed an increased messenger Ribonucleic acid (mRNA) level of late-stage markers such as osteocalcin (OC) and key transcription factor as runt related gene 2 (Runx2). Conclusion: A high hydrophilic scaffold may speed up the osteogenic differentiation for bone tissue engineering.

## 1. Introduction

The recent development of bone tissue engineering (BTE) has brought a new alternative for the treatment of bone defects, especially larger bone defects with various causes [1]. The main determinants of successful bone regeneration are the selection of the materials and the design of the scaffold [2]. Degradable polymers have been applied as promising bone defect scaffold materials such as polyglycolide (PGA), poly(l-lactide) (PLLA), poly(ε-caprolactone) (PCL), and their copolymers with their capacity of good mechanical properties such as proper E-modulus, absorbability, and the potential to enhance cell proliferation and differentiation. Scaffolds produced by those polymers and copolymers are tunable, e.g., by varying the degradation rate by changing molecular weight and copolymer composition [2]. Thus, their flexibilities to develop customized materials for BTE are needed.

In our previous studies, copolymer scaffolds have consisted of l-lactide and 1, 5-dioxepan-2-one (poly(LLA-*co*-DXO)) or ε-caprolactone (poly(LLA-*co*-CL)), which have been proven as promising candidates for BTE, as compared with (PLLA) and PCL [3,4,5,6]. Short and long term degradation tests of the copolymers (poly(LLA-*co*-DXO)) and (poly(LLA-*co*-CL)) have been investigated, thus proving controllable slower degradation rate in vivo rather than in vitro tests and also indicating their potential usage in tailor-made clinical applications [7,8]. However, the material still needs further improvement in hydrophilicity. We applied different ratios of surfactant Polysorbate 80 (Tween^®^ 80) to improve the hydrophilicity of scaffolds. Though Tween 80 has the high ability to improve the hydrophilicity of polymer materials [9], it is not clear whether it would improve the surface properties of scaffolds for BTE.

Human bone marrow mesenchymal stem cells (hMSCs) are common sources of seeding cells for BTE that are more relevant to reach clinical application. As is known, hMSCs are a rare cell population that are often located in the bone marrow or adipose tissue. An improved cell interaction between seeding cells and scaffold will greatly increase the efficiency of stem cell therapy, especially for the BTE field. Several recent studies have indicated that altered surface properties could stimulate hMSCs for osteogenic differentiation via nano-structure modification [10,11,12], surface coating [13], or topographical features [14]. The utilization of small amounts of surfactant on scaffold materials could achieve significant increases of hydrophilicity. As determined by human like-osteoblast cells, we have successfully gotten corresponding effects of the scaffolds [15]. To verify the application for future clinical use, it was necessary to test the influence of hydrophilicity towards the attachment, proliferation, and differentiation of hMSCs.

In this study, different concentrations of Tween 80 were used as surfactants to tune the hydrophilicities of two copolymer scaffolds. Tween 80 is a non-ionic surfactant and emulsifier that has been widely used as an additive in the food and pharmaceutical industries [9]. Its structure includes three hydrophilic polyethylene glycol side chains with 20 ethylene oxide units that make it suitable as a hydrophilic additive compound. Besides its hydrophilic arms, the long hydrophobic chain is fitted with the polymer’s hydrophobic main chain which could prevent being rinsed away during the salt leaching process. Consequently, the surface hydrophobic property of two copolymers could be changed.

Thus, the aims of the present in vitro study were to enhance the hydrophilicity of the copolymers poly(LLA-*co*-DXO) and poly(LLA-*co*-CL) scaffolds by adding different percentages of Tween 80 and to test the cellular response of hMSCs with different levels of hydrophilicity.

## 2. Materials and Method

### 2.1. Fabrication of Scaffolds

The random copolymers poly(l-lactide-*co*-1,5-dioxepan-2-one) (poly(LLA-*co*-DXO)) and poly(l-lactide)-*co*-(ε-caprolactone) (poly(LLA-*co*-CL)) were synthesized via a previous described method [3]. ε-Caprolactone (Sigma-Aldrich, Germany) was purified by vacuum distillation over CaH_2_, l-lactide (Sigma-Aldrich, Germany) was recrystallized three times from toluene, and 1,5-dioxepan-2-one was synthesized and purified according to earlier described method [16]. The composition of the copolymers was 75% l-lactide (LLA) and 25% of either 1, 5-dioxepan-2-one (DXO) or caprolactone (CL) [17]. In the current study, the DXO refers to (poly(LLA-*co*-DXO)), and CL refers to (poly(LLA-*co*-CL)). The bulk polymerizations were performed 110 °C for 72 h, and Sn(Oct)_2_ was used as catalyst with a monomer:Sn(Oct)_2_ ratio of 10000:1. The amount of Tween 80 was also calculated by comparing the ^1^H-NMR integrations of the corresponding peaks of copolymer (–CH protons on lactide segment, 5.03–5.13 ppm) and Tween 80 (–CH_2_, from the poly(ethylene oxide) segment, 3.65 ppm). Pristine copolymer and Tween 80 (Sigma-Aldrich, Germany) blends were fabricated as films and scaffolds for characterizations. The solid film samples were prepared by solvent casting. In the fabrication of porous scaffolds, sodium chloride was used as a porogen in a leaching method that had been described earlier [3]. The pore size of scaffolds was determined by the size-controlled salt particles that went through sieves and ranged from 90 to 500 μm. Tween 80 was added and blended overnight within the dissolved polymer solution, and three different weight ratios were used: 3%, 10%, and 20%. Eight different materials groups were set up, as presented in Figure 1 and Table 1. When producing scaffolds containing Tween 80 (considering the water solvability of Tween 80), a feeding ratio (data not shown) was used to ensure the Tween 80 composition after salt leaching. A copolymer without Tween 80 was chosen as the control group. After solvent evaporation, scaffold samples were cut into small cylinder pieces of 1 cm in diameter and 1 mm in thickness. Then, salt particles were leached by using deionized water, and the porous scaffolds were then dried in a vacuum. Sterilization was accomplished by electron beam radiation (25 kGy dose) using a pulsed electron accelerator (Mikrotron, Acceleratorteknik, Stockholm) under an inert environment at 6.5 MeV.

### 2.2. Test of Contact Angle and Hydrophilicity after Surface Modification

A water contact angle test was completed on film samples using a CAM 200 contact angle system (KSV instruments Ltd., Finland). A 5 μL drop of Milli-Q water was added on to the surface of the samples, and the measurements were recorded with an optical camera. The average angles were calculated from 5 different locations. The KSV software was used to analyze the frames and measurements. To test the effect of the hydrophilicity of the Tween 80-modified materials, both scaffolds and films were tested upon direct response of water solution. In addition, 50 µl of red color water solutions were applied (Ponceau S red). Time lapse videos were taken with a digital camera (Sony Alpha 77, Tokyo, Japan) mounted with a macro lens (Sony Macro f2.8/50, Tokyo, Japan).

### 2.3. Cytotoxicity Testing by HaCaT Cells

The aim was to test if cytotoxic substances can be extracted from the materials immersed in a cell culture medium. The cytotoxicity testing was performed according to Deutsches Institut für Normung (DIN) International Organization for Standardization (ISO) 10993-5. Different groups’ scaffolds were soaked with Dulbecco’s Modified Eagle Medium (DMEM) (Sigma-Aldrich^®^, Germany) without serum. The medium was collected after 24 ± 2 h, and 10% fetal bovine serum (FBS) was added and pre-warmed in a water bath before using. A fresh DMEM medium with 10% FBS was used as a negative control, whereas a fresh DMEM medium supplemented with 0.1% SDS was used as a positive control.

The HaCaT cell line (DKFZ, Heidelberg, Germany) was used to test the cytotoxicity of the materials. The HaCaT cells were seeded into a 96-well plate with 2 × 105 cells/well in 200 µl of a culture medium. The cells were incubated for 24 ± 2 h in a normal DMEM medium before being changed to extracted media. After being incubated in extract media for 24 ± 2 h, a WST-1 reagent (2-(4-iodophenyl)-3-(4-nitrophenyl)-5-(2,4-disulfophenyl)-2H-tetrazolium) (Roche Molecular Biochemicals, Switzerland) was applied to test cell viability. Samples were incubated in a medium containing WST-1 reagents at 37 °C for 1h. The absorbance at 450 nm was obtained by a micro plate reader (BMG LABTECH, Germany).

### 2.4. hMSC Culture and Seeding for Determination of Attachment, Spreading, Proliferation and Differentiation

To test the cell response on modified scaffolds, hMSCs (StemCell Technologies, Vancouver, BC, Canada) of a passage between 4 and 8 were applied. The cells were cultured in a MesenCult^®^ complete medium (StemCell Technologies), and a MesenCult™ Osteogenic Differentiation Medium was used to test osteogenic differentiation. hMSCs were pre-characterized by a flow cytometry, which showed that >90% of the cells could express the cluster of differentiation (CD)29, CD44, CD105, and CD166. For cell seeding, 2 × 10^5^ of cells in 500 µl of a culture medium were added on top of the scaffold (n = 4 in each group). The cells/scaffold constructs were harvested at 1 h after seeding and rinsed in phosphate-buffered saline (PBS), and a WST-1 assay was used to determine the initial attachment.

To test cell spreading on different materials, the eight groups’ materials listed in Figure 1 were fabricated into films; 3 × 10^4^ hMSCs were seeded on top of each film and then incubated and harvested for 1, 3, and 5 h. The specimens were briefly rinsed in PBS, fixed in 4% paraformaldehyde, and then stained with fluorescein isothiocyanate conjugated CD90 (FITC-CD90) and 4′,6-diamidino-2-phenylindole (DAPI). Images were taken by a fluorescence microscope (Nikon 80i, Tokyo, Japan) and analyzed using the NIS-Elements 3.07^®^ software (Nikon, Tokyo, Japan). Tree samples from each group were analyzed. The average cellular surface area was calculated.

To determine the proliferation of hMSCs, scaffolds from the eight groups were pre-wetted with the MesenCult^®^ complete medium and then seeded with 1 × 10^5^ hMSCs per scaffold. The cells/scaffold constructs were cultured in a static culture condition in culture plates overnight and then transferred to spinner flasks that had been described in a previous study [18]. The constructs were cultured for 3 or 7 days before being retrieved, according to the WST-1 assay. Scaffolds were rinsed in PBS and moved to a medium containing the WST-1 reagent and incubated for 1 h before absorbance was measured, as mentioned before.

The osteogenic medium was applied to induce osteogenic differentiation for 2 weeks, and osteogenic markers were evaluated by real time RT-PCR.

### 2.5. Scanning Electron Microscopy (SEM)

To investigate cellular morphology, 1 × 10^5^ hMSCs were seeded per scaffold, and samples were prepared for SEM following protocol to experience surface observation after 3 h of static incubation or after 3 days in spinner flasks. The protocol was described in a previous study [19]. Briefly, the specimens were firstly placed in α-minimum essential medium (α-MEM) containing 2.5% glutaraldehyde without serum and fixed for 30 min. Then, 2.5% glutaraldehyde was fixed in 0.1 M sucrose and 0.1 M sodium–cacao salt for 30 min, 1% osmium tetroxide was treated for 1 h, and ethanol was classified to 100% dehydration. Critical point drying was carried out when the specimen was mounted on the aluminum scaffold and coated with 10 nm gold platinum. The samples were examined via SEM (JSM 7400F Jeol, Japan), and a voltage of 10 kV was applied.

### 2.6. Osteogenic Differentiation Evaluation by Real-Time RT-PCR

Based on the proliferation and cytotoxicity test results, four groups—DXO, DXO plus 3% Tween, CL, and CL plus 3% Tween—were preceded for bone marker expression studies. Constructs were cultured in spinner flasks for 1 and 2 weeks before harvest. Total RNA was isolated and purified using the Maxwell^®^ 16 RNA purification kit (Promega Corporation, US). RNA purity and quantification were determined by a Nanodrop Spectrophotometer. The procedure was similar to that of a previous study [19]. The reverse transcription reactions were performed with a High Capacity complementary deoxyribonucleic acid (cDNA) Archive Kit (Applied Biosystems^TM^): 1000 ng of total RNA dissolved in 50 µl of nuclease-free water were collected and mixed with a reverse transcriptase (RT) buffer, random primers, deoxy-ribonucleoside triphosphate (dNTPs), and MultiScribe RT. Real time RT-PCR was run under standard enzyme and cycling conditions on a StepOne™ real time PCR system using TaqMan^®^ gene expression assays (Applied Biosystems™): alkaline phosphatase (ALP); collagen type 1 (Col 1), runt related gene 2 (Runx2), bone morphogenetic protein 2 (BMP-2), Osterix (Osx), Osteocalcin (OC), and the Taqman^®^ Pre-Developed Assay glyceraldehyde-3-phosphate dehydrogenase (GAPDH). cDNA corresponding to 10 ng of messenger RNA (mRNA) was used in each PCR reaction, and mixtures were made up in 10 µl triplicates for each target cDNA. Amplification was performed in 96-well thermal cycler plates with a StepOne™ real time PCR system. The results were analyzed by a comparative Ct (cycle threshold) method by StepOne^TM^ software.

### 2.7. Statistical Analysis

The experiment was repeated by using hMSCs from two different donors. SigmaStat 3.1 was applied for statistical processing and analysis. A one-way ANOVA test was performed as the analysis, and significant differences between means were set as *p* < 0.05. All values in bar charts are expressed as mean ± standard deviation.

## 3. Results

### 3.1. Tween 80 Enhances the Hydrophilicity of Materials

A contact angle test was used to measure the surface hydrophilicity of the solid film scaffolds. Each value was calculated from five repeat measurements, and the average value and standard division were recorded; the results are presented in Figure 1. The results of the contact angle test indicated a significant decrease in hydrophobicity by adding Tween 80. With 3% Tween 80, the contact angle of CL was dramatically reduced by 60% from 86° to 34° and by 70% for DXO from 74° to 18°. However, the 10% and 20% Tween 80 had similar effects on both scaffolds and turned hydrophilic with very low contact angles of about 13°.

By taking time lapse videos under the same exposure parameters, the direct response of materials (both porous scaffolds and solid films) to the dye solution was visualized at different time points, as shown in Figure 2. Firstly, all the Tween 80-treated groups showed an improvement of wettability compared to the control, and the liquid could spread immediately into the porous scaffolds within 1 min. There were slight differences in the 3% Tween 80 groups in the beginning, but after 10 min, no visible differences were found between all the Tween 80-treated groups. Secondly, the DXO group showed a higher wettability compared to the CL group. The water could be absorbed after around 5 min for DXO, while it remained round-shaped on the top of the CL scaffolds.

### 3.2. Poly(LLA-co-CL) and Poly(LLA-co-DXO) with 3% Tween 80 Have No Cytotoxicity

The readings of the WST-1 results of all the groups were compared with the control group to find a relative percentage to indicate cytotoxicity. The scale of cytotoxicity was classified as follows: 80-100% proliferation relative to the control was considered non cytotoxic, 70–80% as weakly cytotoxic, 60–70% as moderately cytotoxic, and 0–60% as strongly cytotoxic. The results are summarized in Table 2 and indicated a non-cytotoxic effect among the DXO, CL, DXO plus 3% Tween, and CL plus 3% Tween groups. Interestingly, there was higher proliferation on the 3% Tween groups than the DXO and CL control groups, which indicated a stimulation effect of 3% Tween 80 on the viability of the HaCaT cells. While the DXO plus 10% Tween and CL plus 10% Tween groups were moderately cytotoxic, the DXO plus 20% Tween and CL plus 20% Tween scaffolds were strongly cytotoxic to HaCaT cells (Table 2).

### 3.3. Tween 80 Delayed Spreading of hMSC

Cell attachment on porous scaffolds: The WST-1 results (Figure 3A) after one hour indicated the initial attachment for hMSCs on the different scaffolds. Unattached cells were gently rinsed away before the WST-1 reagent was added, so only the attached cells would react with the WST-1. The results showed more cells attached to the surface of the DXO, DXO plus 3% Tween, and CL groups in comparison to the other groups. This demonstrated that 10% and 20% Tween could reduce hMSC’s initial attachments compared to the 3% Tween and control groups.

During the multiple time points’ observation of hMSC spreading on the surface of films, after one, three, and five hours, the average cell surface areas were analyzed by using image analysis software. The results from image analysis demonstrated a better spreading on the films from the control groups (DXO and CL) than the Tween-added groups (Figure 3B and Figure 4).

It was found that 3% Tween 80 with poly(LLA-*co*-DXO) had no negative effect on the proliferation of MSCs after performing dynamic cultures of hMSCs with different scaffolds in spinner flasks, the WST-1 assay was applied to determine cellular proliferation. Generally, there were higher readings in the control groups (DXO and CL), and no significant difference was shown between the DXO and DXO plus 3% Tween groups. There was lower reading from the CL plus 3% Tween group than the CL group after 3 and 7 days. It was very clear that there were significant lower absorbance values from the 10% and 20% Tween-added groups compared with the control groups after 3 and 7 days (Figure 3A).

### 3.4. Scanning Electron Microscopy (SEM) Reveals 3% Tween 80-Added Groups Have Better Spreading and Growth

To illustrate the cellular morphology of attachment and proliferation, SEM images were taken after three hours and three days. After three hours, the better spreading of hMSCs from DXO and CL groups was observed in comparison to the Tween 80-added groups. Fewer cells could spread in the 10% Tween groups, and almost no cells could spread after three hours from the 20% Tween 80 groups. After 3 days, hMSCs spread and grew in the control groups and in the groups with 3% and 10% Tween 80. However, there were fewer cells observed on the samples with 20% Tween (Figure 5).

### 3.5. Poly(LLA-co-CL) with 3% Tween 80 Enhance the Expression of RunX2 and OC at Week 2

Based on the cytotoxic results and cell proliferation results, only the control groups (DXO and CL groups) and two modified groups (DXO plus 3% Tween and CL plus 3% Tween) were selected and proceeded to the real time RT-PCR assay to evaluate their effect on osteogenic differentiation for up to two weeks. Generally, there was no significant difference among groups after one week, and most of the osteogenic markers increased at week two. There was a significant higher expression of RunX2 and OC in the CL plus 3% Tween group compared to the other groups (Figure 6).

## 4. Discussion

Our previous in vitro studies indicated that, in comparison with PLLA, seeding hMSCs or human osteoblast-like cells (HOB) on either poly(LLA-*co*-DXO) or poly(LLA-*co*-CL) scaffolds would improve their attachment, proliferation, and differentiation [8,20]. Poly(LLA-*co*-DXO) scaffolds had also been applied to bone regeneration in a rat calvarias defect model [18]. Previous studies have suggested that these materials have great potential for bone tissue engineering.

Poly(LLA-*co*-DXO) and poly(LLA-*co*-CL) are composed of 75% LLA and 25% CL or DXO. Both their mechanical properties and degradation rates have been found to perform better than PLLA and PCL [3]. Based on our previous findings, we have been continuously pursuing methods to improve different properties of the materials. As one of the most important properties for biological materials and scaffolds, hydrophilicity was the main focus point in this study. We used Tween 80 to tune the hydrophilicity of both films and 3D porous scaffolds in order to find the potentially optimal hydrophilicity of poly(LLA-*co*-DXO) and poly(LLA-*co*-CL).

The aim to enhance the hydrophilicity was successfully fulfilled in this study. The contact angle was measured with the materials produced in film shape. By adding only 3% of Tween 80, the contact angle significantly decreased from above 70° to below 40°. These results confirmed that only a very small amount of Tween 80 was needed to obtain a high hydrophilicity. Though contact angle measurements of porous scaffolds might be adequate, the contact that happened at the interface of water drop and the scaffold surface required sufficient interactions. However, for the porous scaffold with the bumpy surface morphology, it was hard to observe the contact at the interface, and this will limit the accuracy of the measurement. To correspond to the contact angle’s results of films, a liquid absorption test was implemented with both scaffold and film materials. The rapid absorption of liquid was observed in most groups with Tween 80. For the scaffolds, we assumed that 3D scaffolds had the same degree of hydrophilicity with corresponding film samples when they had similar compositions. Our previous experiments showed that 0.5% and 1% of Tween 80 had little effect on the hydrophilicity of the materials [9], as well as that the higher the concentration of Tween 80 was, the higher the hydrophilicity of the scaffold materials was [15]. When hydrophilicity is too high, it can also prevent cell adhesion, such as when polyethylene glycol (PEG) is used for coatings that prevent protein adsorption. Therefore, in this experiment, 3%, 10%, and 20% Tween 80 were selected to study the hydrophilic changes of scaffolds. All scaffolds were horizontally segmented and an ^1^H-NMR assay showed that only a small amount of Tween 80 was required to obtain a highly hydrophilic surface. Thus the addition of 3% Tween 80 was enough to improve the surface hydrophilicity of the scaffolds.

For potential clinical applications, the safety of the modified materials was evaluated based on the ISO standard procedures. We selected HaCaT cells rather than hMSCs for the cytotoxicity evaluation because HaCaT cells could keep their properties and have a better biological response. Toxicity effects of 10% and 20% Tween 80 on the sensitive HaCaT cell line were detected. This result was consistent with the proliferation results obtained with the WST-1 assay from 10% or 20% Tween 80 groups. From the cell morphology obtained through SEM, hMSCs could still grow on the scaffolds with 10% Tween 80; however, the cellular activity was significantly reduced according to the results of the WST-1 assay.

The next important issue was to evaluate the initial cell attachment, spreading, proliferation, and differentiation of hMSCs on the modified copolymers. Cell attachment is influenced by several material properties, e.g., surface topography, surface charge, porosity, degradation, and the expression of cell surface molecules and their interaction with the material [10,13,14,21]. In this study, the modification did not lead to the expected enhancement of cell adhesion, and this might have been due to both the property of the cell’s phenotype and its degree of hydrophilicity. Several studies were performed regarding materials with gradient hydrophilicity prepared by different methods and different cell types [22,23,24]. Results indicated that the greater adhesion and growth of cells obtained from samples with a contact angle of 55° in different cell types. In the same trend in another study, a moderately hydrophilic surface was also observed for high cell adhesion [25]. There were studies describing the differences of cell spreading on gradient hydrophilic sheets in a medium with or without serum. Results suggested that cell adhesion has an essential relationship with protein resorption in serum. Kennedy et al. reported the differences of cell spreading on fibronectin-mediated gradient sample sheets [24]. Each protein has unique active sites that appear with different surface environments, and some proteins are independent with surface hydrophilicity [26,27]. Moreover, the protein on the surface can change and vary along with adsorption time [28]. Some researchers have suggested that the influence of hydrophilicity is weak on cell initial adhesion, whereas it is significantly related to cell proliferation. Another important reason for the different results could be the cell phenotypes. In this study, the influence of hydrophilicity could be addressed for hMSCs that might be more sensitive to high hydrophilicity or protein resorption. In the current study, a higher hydrophilic scaffold showed a slower attachment and delayed spreading of hMSCs. This indicated that high hydrophilicity might influence the initial activities of hMSCs. We speculate that Tween 80, as a surfactant, altered the surface protein properties of the material rather than just improve the hydrophilicity, thereby affecting the initial adhesion and proliferation.

Osteogenic differentiation was evaluated by the specific markers such as ALP, Col1, BMP2, Osterix, Runx2, and OC, which is especially meaningful because hMSCs are a very important cell source for future clinical applications. Regarding the expression of osteogenesis-related mRNA, increased expressions were noticed from the second week compared to the first week for almost all markers, which indicated that all scaffolds would have a positive influence on the differentiation of hMSCs in dynamic culture environments. A tendency of higher expression of differentiation markers like Osterix, ALP, and BMP-2 could be observed from the mean value after two weeks from the DXO plus 3% Tween and CL plus 3% Tween groups, though this difference was not statistically significant. There is always a balance between cellular proliferation and differentiation. It is logical that there was a relatively lower proliferation rate from the CL plus 3% Tween group when we got the result that the key transcription factor Runx2 and the late phase differentiation marker OC from the CL plus 3% Tween group had significantly higher expressions. These results suggested that cells reached a relatively later stage of differentiation, which could be positive for potential clinical applications. The variances between the DXO plus 3% Tween and CL plus 3% Tween groups could be due to the different binding ability of Tween 80 for different copolymers.

In the RT-PCR analysis, though some results showed no statistical differences, there were obviously higher expressions of ALP, BMP2, and Osterix in the 3% Tween groups, which might have been due to standard deviation. Further studies may consider increasing sample sizes. That may mean that when modified with 3% Tween 80, hMSCs are able to have osteogenesis differentiation after two weeks. Runx2 is an important transcription factor in bone development and plays an important role in regulating the differentiation of bone forming cells. The significant increase of RunX2 and OC indicated a promotion of osteogenic effect in the CL plus 3% Tween group. This was consistent with our previous in vivo experiments in rats [9]. After the modification with 3% Tween, both CL and DXO had similar properties of surface hydrophilicity and cell attachment, while the osteogenic differentiation had a different effect. This could indicate that the different binding sites of two copolymers might stimulate different cellular signals for hMSC, this warrants a further study to clarify its mechanism.

Other researchers have fabricated poly(DL-lactic-*co*-glycolic acid) (PLGA)/Tween 80 films and porous scaffolds to test the hydrophilicity and cell compatibility of the materials [29]. The addition of 10% Tween 80 to PLGA enhanced the cell attachment of chondrocytes and cell compatibility in comparison to the PLGA scaffold. Generally, the results of this study were not totally in consistent with previous studies. Several factors were variant: Firstly, different copolymer materials were applied, this might lead to different cell behaviors. Secondly, different cell phenotypes could react differently, so the results of this study suggested that hMSCs might be more sensitive to highly hydrophilic surfaces. Moreover, in this study, dynamic culture conditions were applied under different culturing times. A further future step could be the verification of specific protein resorption on different hydrophilic scaffolds. Regarding the different sources of MSCs, mandible mesenchymal stem cells (M-MSCs) could be included in the future study because they have been investigated as promising candidates for craniofacial bone tissue engineering [30].

## 5. Conclusions

This work found a tailored wettability scaffold that may enhance efficiency in bone tissue engineering. The hydrophilicity of poly(LLA-*co*-DXO) and poly(LLA-*co*-CL) could be significantly increased by blending a low ratio of Tween 80. The increased hydrophilicity by 3% Tween 80 showed clear stimulation on the osteogenic differentiation of hMSCs. However, there was no significant increase on the initial cell attachment and spreading after modification. The proliferation of hMSCs was not obviously enhanced on the Tween 80-added scaffolds. The differentiation of hMSCs was stimulated, and several osteogenesis-related genes, such as OC and Runx2, were up-regulated by adding 3% Tween 80.

## Figures and Tables

**Figure 1 polymers-12-01453-f001:**
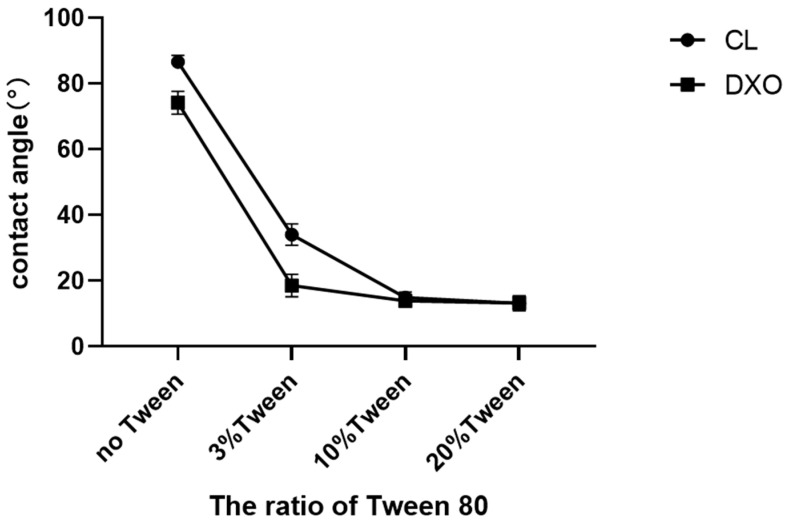
3%, 10% or 20% Tween 80 decreased the contact angle of poly(LLA-*co*-CL) and poly(LLA-*co*-DXO), implying that all of the three ratios of Tween 80 could improve the hydrophilicity of the copolymers (n = 5).

**Figure 2 polymers-12-01453-f002:**
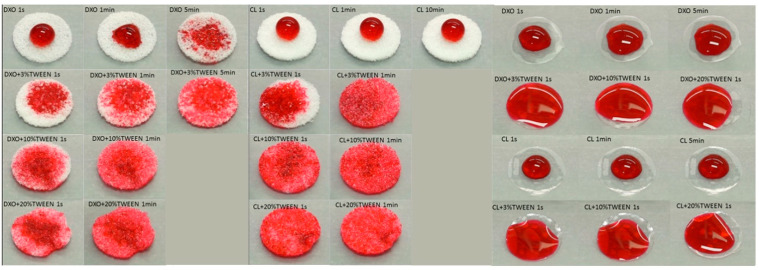
The spreading of Ponceau S Red solution on the films (the right) and the penetration of Ponceau S red solution into different scaffolds (the left).The dye solution could not totally spread on the films of the DXO and CL groups, but it could quickly diffuse on the films that contained Tween 80. Meanwhile, DXO showed a better spreading than CL.

**Figure 3 polymers-12-01453-f003:**
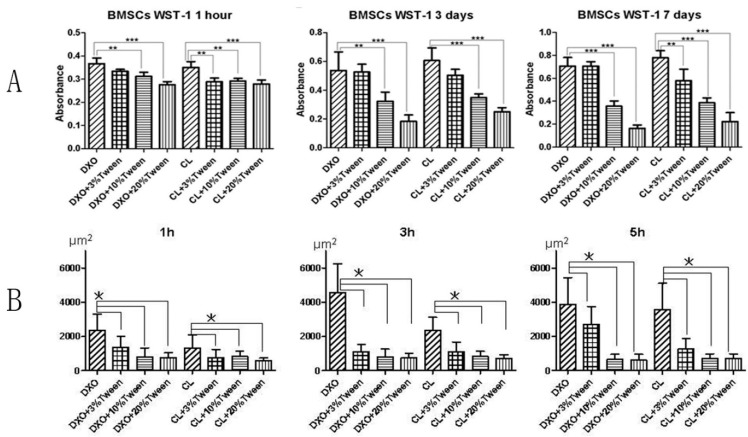
(**A**). WST-1 results after 1 h, 3 days, and 7 days. (** *p* < 0.05, *** *p* < 0.01). 3% Tween 80 in DXO had no negative effect on the initial attachment and proliferation of cells. (**B**). Average cell areas of hMSCs on films from different groups after 1, 3, and 5 h (* *p* < 0.05).

**Figure 4 polymers-12-01453-f004:**
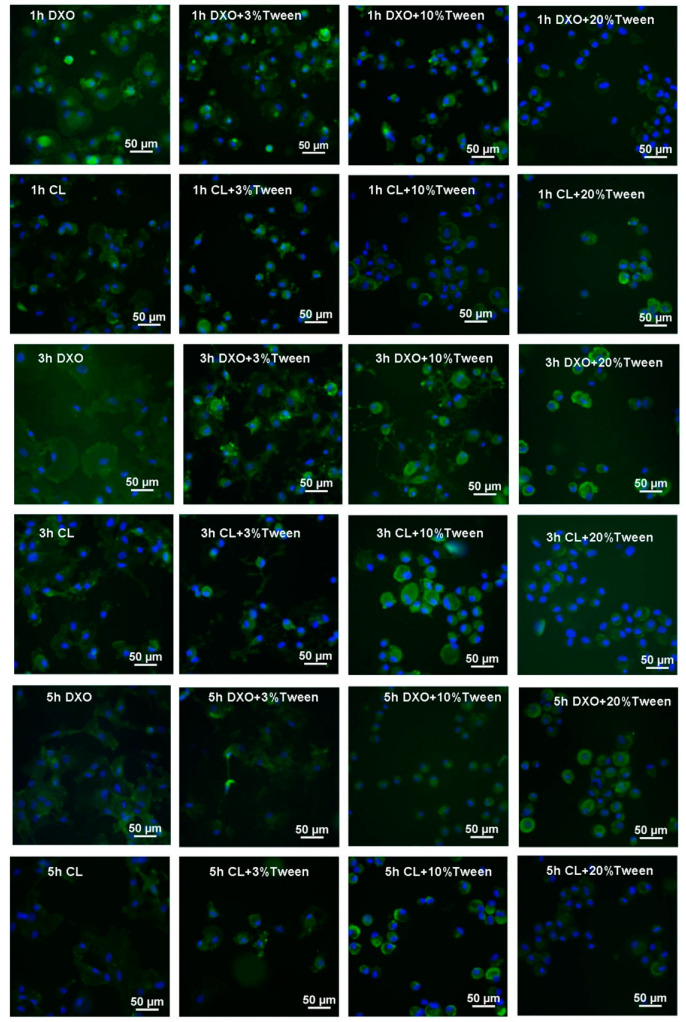
Spreading of human bone marrow mesenchymal stem cells (hMSCs) on membranes of different groups from 1 to 5 h. hMSCs were stained with fluorescein isothiocyanate conjugated cluster of differentiation 90 (FITC-CD90) (green) and all nucleus with 4′,6-diamidino-2-phenylindole (DAPI) (blue).

**Figure 5 polymers-12-01453-f005:**
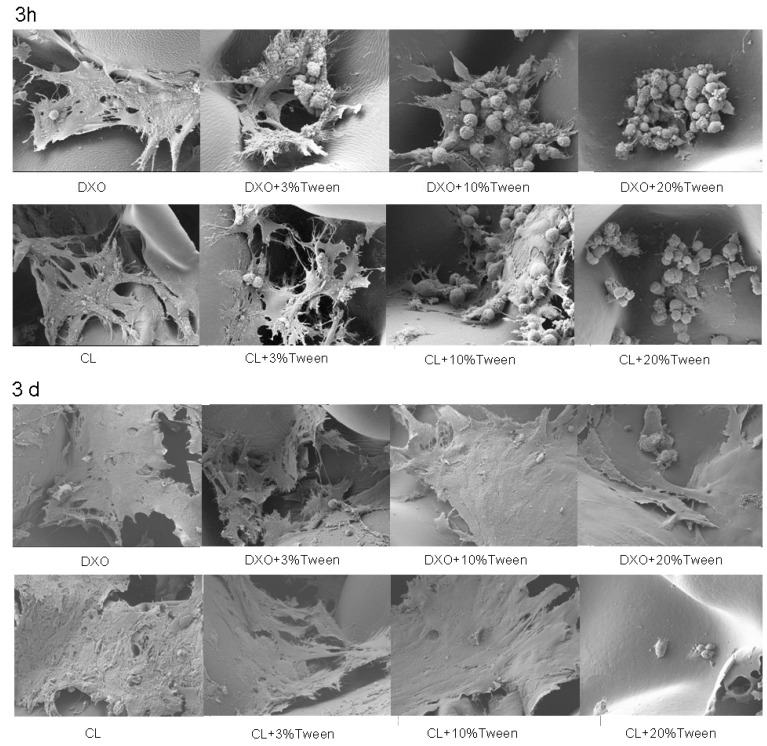
SEM results after 3 h and 3 days. Results from 3 h showed the better spreading of hMSCs on control groups compared with the Tween 80-added groups. Three days’ results showed that most cells could spread and grow from the control groups and the 3% and 10% Tween 80-added groups. Fewer cells could be observed in the 20% Tween 80-added groups.

**Figure 6 polymers-12-01453-f006:**
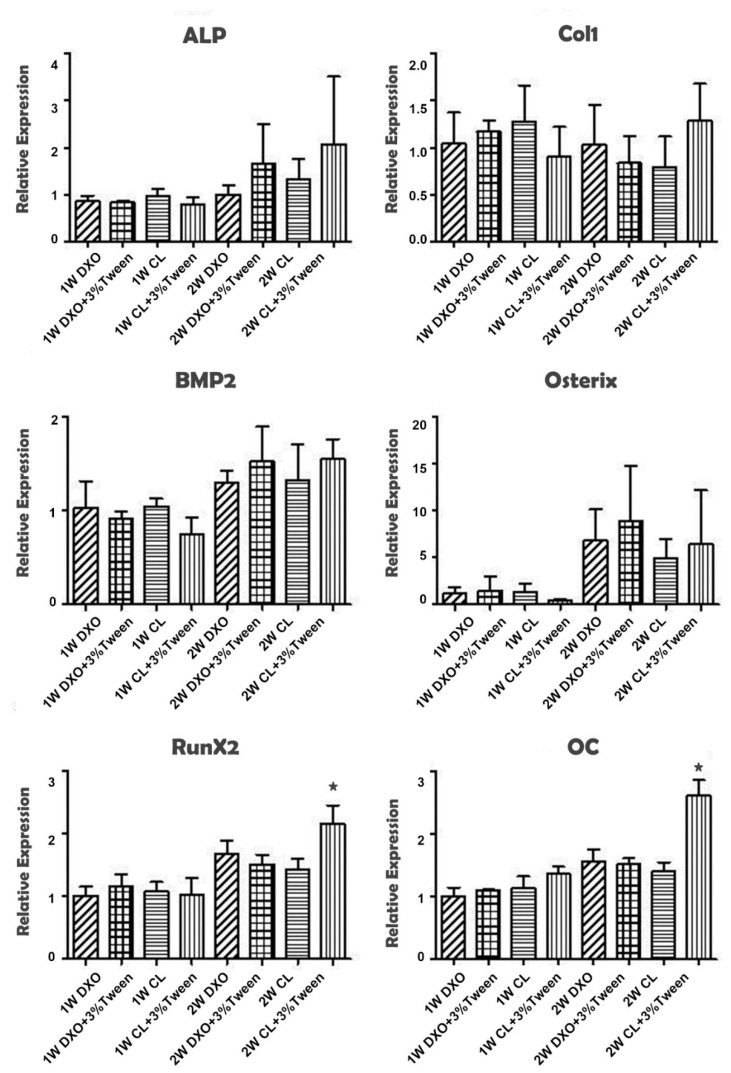
Quantitative real-time reverse transcriptase (RT)-PCR analysis from DXO, CL, DXO plus 3% Tween, and CL plus 3% Tween groups after culturing for 1 and 2 weeks. Runt related gene 2 (Runx2) and osteocalcin (OC) showed significant higher expression from the CL plus 3% Tween group after 2 weeks’ culture (* *p* < 0.05).

**Table 1 polymers-12-01453-t001:** The water contact angle results measured on the film samples.

	Value	SD
CL	86.65	1.94
CL + 3%Tween	34.04 *	3.23
CL + 10%Tween	14.87 *	1.71
CL + 20%Tween	13.15 *	2.28
DXO	74.19	3.45
DXO +3%Tween	18.54 #	3.40
DXO + 10%Tween	13.89 #	1.70
DXO + 20%Tween	13.21 #	1.93

* Refers to significant differences with CL (which refers to poly(L-lactide-*co*-ε-caprolactone), also known as poly(LLA-*co*-CL)), and # refers to significant differences with DXO (which refers to poly(LLA-*co*-DXO)), also known as poly(l-lactide-*co*-1,5-dioxepan-2-one)).

**Table 2 polymers-12-01453-t002:** The cytotoxicity results measured by the WST-1 (2-(4-iodophenyl)-3-(4-nitrophenyl)-5-(2,4-disulfophenyl)-2H-tetrazolium) assay.

Groups	Mean (%)	SD(%)	Cytotoxic Potential *
DXO	87.6	4.18	non cytotoxic
DXO plus 3% Tween	103.6	4.91	non cytotoxic
DXO plus 10% Tween	66.8	9.59	moderately cytotoxic
DXO plus 20% Tween	41.8	8.41	strongly cytotoxic
CL	86.3	8.21	non cytotoxic
CL plus 3% Tween	99.8	6.08	non cytotoxic
CL plus 10% Tween	65.8	8.60	moderately cytotoxic
CL plus 20% Tween	52.3	10.32	strongly cytotoxic

DXO, CL, and DXO/CL plus 3% Tween 80 groups revealed non cytotoxic potential. * 80–100% proliferation relative to the control was considered non cytotoxic, 70–80% weakly cytotoxic,60–70% moderately cytotoxic, and 0–60% strongly cytotoxic.

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
