# Peer review of "Altered Surface Hydrophilicity on Copolymer Scaffolds Stimulate the Osteogenic Differentiation of Human Mesenchymal Stem Cells"

_polymers, 2020, doi:10.3390/polym12071453_

Round 1

Reviewer 1 Report

‘Altered surface hydrophilicity on copolymer scaffolds stimulate human mesenchymal stem cells cell proliferation and osteogenic differentiation’

This work is interesting but it needs major revision for publication, as commented below:

1. Authors need to separate the effects of Tween surfactant on cell toxicity; have to do indirect assay using insert membrane.

2. The assays on cell differentiation should also be conducted by adding Tween to the raw polymers (w/o Tween).

3. Gene expression data is not enough to support their hypothesis; authors need to do Alizarin red staining of cells to see the cells are really osteogenic.

4. Many recent works on stimulating MSCs functions for osteogenic differentiation need to be referenced in order to feedback the recent trend in this area, as shown below:

- Waddell SJ et al. Biomimetic oyster shell–replicated topography alters the behaviour of human skeletal stem cells. Journal of Tissue Engineering, 2018; 9: 2041731418794007  / Kang ES et al. Guiding osteogenesis of mesenchymal stem cells using carbon-based nanomaterials, Nano Converg, 2017, 4  /  Lee DJ et al. Osteogenic potential of mesenchymal stem cells from rat mandible to regenerate critical sized calvarial defect. J. Tissue Eng., 2019; 10: 2041731419830427  / Hwang J, et al. Artificial cellular nano-environment composed of collagen-based nanofilm promotes osteogenic differentiation of mesenchymal stem cells, Acta Biomater, 2019, 86. /   Federica Re et al. 3D gelatin-chitosan hybrid hydrogels combined with human platelet lysate highly support human mesenchymal stem cell proliferation and osteogenic differentiation. J. Tissue Eng. 2019; 10: 2041731419845852.  / Zhu L, et al. Effect of the nano/microscale structure of biomaterial scaffolds on bone regeneration, Int J Oral Sci, 2020, 11.

5. Authors should discuss more on the relationship of hydrophilicity and cell adhesion / differentiation.

6. Authors need to do test using other cells if the behavior only happened to MSCs.

Author Response

Response to Reviewer 1 Comments

Dear reviewer,

Thanks very much for constructive comments, please find attached revised manuscript. Suggested parts have been added, rewritten, explained in a better way, the English language has been checked and polished a lot carefully. Please let us know if anything could be improved further and hope this could meet the demands for publishing.

Best wishes,

Zhe Xing

Point 1: Authors need to separate the effects of Tween surfactant on cell toxicity; have to do indirect assay using insert membrane.

Response 1: Cell toxicity effect was tested according to DIN ISO 10993-5 standard, HacaT cells were tested with extracted medium. This procedure has a similar effect as the reviewer suggested to do indirect assay using insert membrane. Thanks for valuable suggestion and we will consider in the further experiments other methods to test cell toxicity.

Additionally, we have tested direct cellular response of osteoblast to culture medium containing different ratios of Tween 80 in previous studies.  (Sun Y*, Xing Z*, et al. Surfactant as a Critical Factor When Tuning the Hydrophilicity in Three-Dimensional Polyester-Based Scaffolds: Impact of Hydrophilicity on Their Mechanical Properties and the Cellular Response of Human Osteoblast-Like Cells. Biomacromolecules. 2014).

Point 2: The assays on cell differentiation should also be conducted by adding Tween to the raw polymers (w/o Tween).

Response 2: Thanks for advices, and we will continue this in our further studies. We have considered this during the design of experiments, but its difficult to keep Tween 80 with materials if they are not integrated into polymers since we need to change medium frequently.  

Point 3: Gene expression data is not enough to support their hypothesis; authors need to do Alizarin red staining of cells to see the cells are really osteogenic.

Response 3: Thanks for advices, and we will include this in our future studies to verify at protein level for stem cells. We did consider this investigation when we designed the experiment, but there is actually also a technically challenging to perform Alizarin red staining on scaffolds with different hydrophilicity because materials will absorb staining reagent quite differently and may result in bias. We consider to verify further in protein level with the other methods, for example, western blot, etc.

From the PCR result, we could see significant higher expression of osteocalcin and Runx2, the other markers (ALP, Col1, BMP2 and Osterix) have higher mean values although the results were not statistically significant. Our previous animal study has also verified in vivo that 3% Tween could enhance bone formation (Mohammed A. Yassin, etal. Surfactant tuning of hydrophilicity of porous degradable copolymer scaffolds promotes cellular proliferation and enhance bone formation. Journal of biomedical materials research A, aug. 2016)

Point 4: Many recent works on stimulating MSCs functions for osteogenic differentiation need to be referenced in order to feedback the recent trend in this area, as shown below:

- Waddell SJ et al. Biomimetic oyster shell–replicated topography alters the behaviour of human skeletal stem cells. Journal of Tissue Engineering, 2018; 9: 2041731418794007  / Kang ES et al. Guiding osteogenesis of mesenchymal stem cells using carbon-based nanomaterials, Nano Converg, 2017, 4  /  Lee DJ et al. Osteogenic potential of mesenchymal stem cells from rat mandible to regenerate critical sized calvarial defect. J. Tissue Eng., 2019; 10: 2041731419830427  / Hwang J, et al. Artificial cellular nano-environment composed of collagen-based nanofilm promotes osteogenic differentiation of mesenchymal stem cells, Acta Biomater, 2019, 86. /   Federica Re et al. 3D gelatin-chitosan hybrid hydrogels combined with human platelet lysate highly support human mesenchymal stem cell proliferation and osteogenic differentiation. J. Tissue Eng. 2019; 10: 2041731419845852.  / Zhu L, et al. Effect of the nano/microscale structure of biomaterial scaffolds on bone regeneration, Int J Oral Sci, 2020, 11.

Response 4: Thanks for suggestion and these references were carefully checked and have been added to the introduction and last part of discussion.

Point 5: Authors should discuss more on the relationship of hydrophilicity and cell adhesion / differentiation.

Response 5: More discussion has been added about this point. It is very interesting in this study that cells had delayed spreading, but enhanced differentiation on a more hydrophilic surface. It is not the highest hydrophilicity are best for the cells, our results indicated that a moderate hydrophilicity is proper that will not influence proliferation and stimulate differentiation.

Point 6: Authors need to do test using other cells if the behavior only happened to MSCs.

Response 6: We have earlier published studies in which we have tested effect of Tween 80 on human osteoblast and rat bone marrow stromal cells, this study we also used HacaT cells, but we focused on human MSCs which will have more clinical meaning for potential applications. Different cell types do behavior differently, we have an impression that a small ratio of Tween 80 could stimulate cell proliferation. But this was not so obvious with MSCs, it looks like MSCs are more sensible to hydrophilic surface, but a promising result could be obtained anyway with a low ratio of Tween 80. We will consider to perform experiments with other cells in the further studies.

Reviewer 2 Report

Comment on the paper “Altered surface hydrophilicity on copolymer scaffolds stimulate human mesenchymal stem cells cell proliferation and osteogenic differentiation”, by Zhe Xing et al. A non-ionic surfactant and emulsifier Polysorbate 80 (Tween80) was used to tune the hydrophilicity of poly(LLA-co-DXO) and poly(LLA-co-CL). The modification was applied in the form of solid scaffold as film and porous scaffold. Then, the effect of hydrophilicity was tested on cell attachment, proliferation and differentiation. The cellular cytotoxicity was investigated. Hydrophilicity was improved using these modifications and 3% of surfactant. The paper is clear and well written. All the conclusions are supported by the data and it can be published after minor revision.

Line 87: how the pore size was evaluated?

Line 185: standard deviation?

Line 190: which percentage is retained for application? It seems that 10% is the best compromise, but in regard with toxicity I guess that only 3% will be applied.

It would be interesting to test the effect of polymers crystallinity that can affect the hydrophilicity and the interaction between Tween80 and the polymers.

Author Response

Response to Reviewer 2 Comments

Point 1: Line 87: how the pore size was evaluated?

Response 1: The scaffolds were produced by salt-leaching methods, the pore-size was determined by the salt crystal sizes. Prior to casting, the salt crystals were first sieved by a 500 µm pore size and then 90 um pore size sieves to get a controlled sizes of salt crystals (90-500 µm). Following this step, the scaffolds by salt-leaching generated porous scaffolds with a range of pore size between 90 to 500 µm. Moreover, the pore-size was controlled and measured by micro CT. This part was rewritten in the manuscript.

Point 2: Line 185: standard deviation?

Response 2: Thanks for comments. This has been corrected in the text.

Point 3: Line 190: which percentage is retained for application? It seems that 10% is the best compromise, but in regard with toxicity I guess that only 3% will be applied.

Response 3: Based on the cytotoxic results and cell proliferation results, we chose modified groups (DXO+3% Tween and CL+3% Tween) to precede for real time RT-PCR assay to evaluate their effect on osteogenic differentiation for up to 2 weeks.

Yes, 10% group has lower contact angle but is not most favored by cell to attach and grow.

Point 4: It would be interesting to test the effect of polymers crystallinity that can affect the hydrophilicity and the interaction between Tween 80 and the polymers.

Response 4: Thanks for your valuable suggestions and due to some limitation, this test can’t add into the current study. But this will be definitely conducted in our future studies.